# Ultra-Broadband Tunable Terahertz Metamaterial Absorber Based on Double-Layer Vanadium Dioxide Square Ring Arrays

**DOI:** 10.3390/mi13050669

**Published:** 2022-04-25

**Authors:** Pengyu Zhang, Guoquan Chen, Zheyu Hou, Yizhuo Zhang, Jian Shen, Chaoyang Li, Maolin Zhao, Zhuozhen Gao, Zhiqi Li, Tingting Tang

**Affiliations:** 1School of Information and Communication Engineering, Hainan University, Haikou 570228, China; pengyuzhang@hainanu.edu.cn (P.Z.); chenguoquan@hainanu.edu.cn (G.C.); houzheyu@hainanu.edu.cn (Z.H.); zhangyizhuo@hainanu.edu.cn (Y.Z.); zhaomaolin@hainanu.edu.cn (M.Z.); gaozhuozhen@hainanu.edu.cn (Z.G.); zhiqili@hainanu.edu.cn (Z.L.); 2State Key Laboratory of Marine Resources Utilization in South China Sea, Hainan University, Haikou 570228, China; 3Information Materials and Device Applications Key Laboratory of Sichuan Provincial Universities, Chengdu University of Information Technology, Chengdu 610225, China; tangtingting@cuit.edu.cn

**Keywords:** absorber, metamaterial, vanadium dioxide, terahertz

## Abstract

Based on the phase transition of vanadium dioxide(VO_2_), an ultra-broadband tunable terahertz metamaterial absorber is proposed. The absorber consists of bilayer VO_2_ square ring arrays with different sizes, which are completely wrapped in Topas and placed on gold substrate. The simulation results show that the absorption greater than 90% has frequencies ranging from 1.63 THz to 12.39 THz, which provides an absorption frequency bandwidth of 10.76 THz, and a relative bandwidth of 153.5%. By changing the electrical conductivity of VO_2_, the absorption intensity can be dynamically adjusted between 4.4% and 99.9%. The physical mechanism of complete absorption is elucidated by the impedance matching theory and field distribution. The proposed absorber has demonstrated its properties of polarization insensitivity and wide-angle absorption, and therefore has a variety of application prospects in the terahertz range, such as stealth, modulation, and sensing.

## 1. Introduction

Metamaterials, also referred to as artificial composite materials with special electromagnetic properties, usually consist of sub-wavelength structure arranged in periodic arrays. In recent years, the applications of terahertz waves in communication [1], sensing [2], imaging [3], and other processes have attracted extensive attention. To facilitate the development of terahertz technology, various functional devices based on metamaterials, such as polarization converters [4,5], absorbers [6,7], and metalenses [8,9], have been proposed. Among these devices, the metamaterial perfect absorber has played a significantly important role in the terahertz field due to its ultra-high absorption efficiency, small size, and light weight compared with traditional electromagnetic wave absorbers, and is widely used in sensing [10,11], energy collection [12,13], and stealth technology [14,15].

Since the first metamaterial perfect absorber was proposed by Landy et al. in 2008 [16], various types of narrowband [17,18], multi-band [19,20], and broadband [21,22] absorbers have been proposed. However, the common challenge faced by these absorbers is that once the structural parameters are determined, it is difficult to adjust the electromagnetic response characteristics. In order to achieve reconfigurability, researchers have proposed a variety of tuning methods such as electrical reconfigurability based on diodes [23], transistors [24], liquid crystals [25], graphene [26], and other loads; silicon-based optical reconfigurability [27,28]; and VO_2_-based thermal reconfigurability [29,30]. In addition, there are hybrid metamaterials with multiple controllable materials [31,32]. As a temperature-controlled phase change material, VO_2_ will transform from insulator phase to metal phase when the temperature reaches 340 K. During this process, the lattice structure of VO_2_ will change from monoclinic structure to tetragonal structure, and the electrical conductivity is improved significantly. In the terahertz range, the permittivity of VO_2_ changes rapidly with changes in conductivity [33,34]. Thus, VO_2_-based absorbers are expected to achieve dynamically tunable properties in the terahertz range. Some previously reported broadband tunable absorbers based on VO_2_ are shown in Table 1.

In this paper, we introduce an ultra-broadband tunable terahertz metamaterial absorber consisting of a metal ground layer and two-layer VO_2_ square ring arrays of different sizes embedded in a dielectric. Through the transition between the insulator phase of VO_2_ and the metal phase, the absorption intensity can be continuously tuned. According to impedance matching theory and field distribution, the physical mechanism of complete absorption is analyzed and elucidated. In addition, the absorption properties under different polarization and incident angles are also investigated. Clearly, this ultra-broadband tunable metamaterial absorber has broad application prospects in the terahertz field.

## 2. Design and Simulation

The unit cell of our proposed terahertz absorber is illustrated in Figure 1, which consists of two-layer VO_2_ square ring arrays of different sizes surrounded by Topas with a 0.2-μm thick gold film at the bottom. Figure 1b,c shows the top views of the unit cell with the large square ring and the small square rings, respectively. The period P of the unit cell and the geometric parameters of the VO_2_ square rings are set as P = 30 μm, P_1_ = 26 μm, w_1_ = 7 μm, P_2_ = 9 μm, w_2_ = 6 μm, and d = 6 μm. Figure 1d is the side view of the unit cell, where the thicknesses of the VO_2_ square rings, the spacer layer and the cover layer are selected as t_1_ = 0.1 μm, t_2_ = 0.15 μm, h_1_ = 7 μm, h_2_ = 7 μm, and h_3_ = 6 μm.

The electromagnetic response of the absorber is simulated using the finite element method in commercial software COMSOL Multiphysics. The periodic boundaries are applied in the x and y directions, and perfectly matching layers are set along the z direction. In simulations, the optical properties of VO_2_ in terahertz range can be expressed by the Drude model [30,41,42,43]
(1)$$\varepsilon \left(\omega \right)=\varepsilon_{\infty }-\frac{\omega_{\rho }^{2}\left(\sigma \right)}{\left(\omega^{2}+i\gamma \omega \right)}$$
where *ε*_∞_ = 12 is the permittivity at the infinite frequency and *γ* = 5.75 × 10^13^ rad/s is the collision frequency. The plasma frequency can be written as
(2)$$\omega_{\rho }^{2}\left(\sigma \right)=\frac{\sigma }{\sigma_{0}}\omega_{\rho }^{2}\left(\sigma_{0}\right)$$
with *σ*_0_ = 3 × 10^5^ S/m and *ω_ρ_* (*σ*_0_) = 1.4 × 10^15^ rad/s. When the VO_2_ is in the metal phase and the insulator phase, the electrical conductivity σ is 2 × 10^5^ S/m and 200 S/m, respectively. Similarly, the relative permittivity of gold can also be described by the Drude model [42,43,44]
(3)$$\varepsilon_{Au}=1-\frac{\omega_{\rho }^{2}}{\left(\omega^{2}+i\gamma \omega \right)}$$
where plasma frequency *ω_ρ_* = 1.37 × 10^16^ rad/s and collision frequency *γ* = 1.2 × 10^14^ rad/s. The relative permittivity of Topas is 2.35 [45]. Since the thickness of the gold film is much larger than the skin depth of a typical terahertz wave, transmittance T(*ω*) can be regarded as 0. Therefore, absorptance can be calculated as
(4)$$A\left(\omega \right)=1-R\left(\omega \right)=1-{\left|S_{11}\left(\omega \right)\right|}^{2}$$
where *R*(*ω*) is reflectance and *S*_11_(*ω*) is reflection coefficient.

## 3. Results and Discussions

The absorption spectra under normal incidence are displayed in Figure 2a, when VO_2_ square rings are in the metal phase. The result shows that the absorptance exceeds 90% for frequencies between 1.63 THz and 12.93 THz, and the absorption frequency bandwidth is as high as 10.76 THz. Moreover, there are seven absorption peaks with frequencies located at f_1_ = 1.93 THz, f_2_ = 3.38 THz, f_3_ = 6.76 THz, f_4_ = 7.16 THz, f_5_ = 9.44 THz, f_6_ = 10.27 THz, and f_7_ = 11.74 THz, and their absorptance reaches 99.77%, 99.27%, 99.79%, 98.58%, 97.79%, 99.55%, and 99.18%, respectively. Figure 2b shows the absorption spectra of the absorber at different polarization angles from 0° to 90°. It is obvious that the designed absorber has excellent polarization insensitivity due to the symmetry of the structure [37].

In fact, as a typical phase change material, VO_2_ will change its conductivity by 3–5 orders of magnitude when it is transformed from the insulator phase to the metal phase. The conductivity relationship of VO_2_ corresponding to different temperatures in the process is [46,47]
(5)$$\sigma =-i\varepsilon_{0}^{}\omega \left(\varepsilon_{c}^{}-1\right)$$

Among them, *σ* is the conductivity of the composite system, *ε*_0_ is the dielectric constant of the vacuum, and *ε_c_* is the dielectric function of the composite system which is related to temperature. We plot the relationship between conductivity of VO_2_ and temperature as shown in Figure 3 [46]. As the temperature increases to the phase transition point of 340 K, the conductivity of VO_2_ abruptly changes and tends to stabilize when the temperature reaches 360 K. The whole process is reversible.

The absorption spectra of different conductivities are shown in Figure 4a. When the conductivity changes from 200 S/m to 2 × 10^5^ S/m, the absorptance of the absorber increases continuously from 4.4% to about 99.9%, while all central peaks of absorptance happen at almost the same frequency. The continuously tunable physical mechanism is mainly caused by the change of the permittivity of VO_2_. Figure 4b,c display the real and imaginary parts of the dielectric constant of VO_2_ as a function of conductivity. It can be seen that the imaginary part of the permittivity varies more significantly with the conductivity. Because the real part of the permittivity is related to the resonant frequency and the imaginary part is related to the loss, the position of the absorption bandwidth keeps almost unchanged, while the intensity of the absorption spectrum varies significantly.

The physical mechanism of perfect absorption can be explained by the impedance matching theory. When the terahertz wave is under normal incidence, the absorptance and relative impedance of the absorber can be obtained by [48,49]
(6)$$A\left(\omega \right)=1-R\left(\omega \right)=1-{\left|\frac{Z-Z_{0}}{Z+Z_{0}}\right|}^{2}=1-{\left|\frac{Z_{r}-1}{Z_{r}+1}\right|}^{2}$$
(7)$$Z_{r}\left(\omega \right)=\sqrt{\frac{{\left(1+S_{11}\left(\omega \right)\right)}^{2}-S_{21}{\left(\omega \right)}^{2}}{{\left(1-S_{11}\left(\omega \right)\right)}^{2}-S_{21}{\left(\omega \right)}^{2}}}$$
where *Z*_0_ = 377 Ω is the free space impedance, *Z* stands for the effective impedance of the absorber, and *Z_r_* = *Z*/*Z*_0_ represents the relative impedance. Since the thickness of the gold film is much larger than the skin depth, the transmission coefficient *S*_21_(*ω*) is practically zero, the relative impedance can be simplified as
(8)$$Z_{r}\left(\omega \right)=\left|\frac{1+S_{11}\left(\omega \right)}{1-S_{11}\left(\omega \right)}\right|$$

When the relative impedance *Z_r_* is at 1, the impedance of the absorber matches that of free space. It means that the reflectance of the absorber is minimized and the absorptance is maximized. Figure 5a,b exhibits the real and imaginary parts of the relative impedance *Z_r_* for different conductivities, respectively. As the conductivity increases, the real part of the relative impedance gradually approaches 1, and the imaginary part gradually approaches 0, which proves that the impedance of the absorber gradually matches the impedance of the free space. When the conductivity of VO_2_ is in the metal phase, the designed absorber meets the requirements of impedance matching and can achieve minimal reflection.

To further illustrate the physical mechanism of the designed absorber with high absorptance in the ultra-broadband, Figure 6 and Figure 7 present the power distribution on the xy plane of the resonator at seven absorption peaks. For large-sized square rings, as shown in Figure 6, the energy is mainly concentrated in the gaps of adjacent rings at the resonant frequency f_1_, indicating that the coupling between adjacent unit cells is particularly strong. For the resonant frequencies f_2_, f_4_, and f_5_, the energy is concentrated in a single resonator, while the absorption is still dominated by the coupling effect between adjacent unit cells at f_3_. For frequencies f_6_ and f_7_, the energy is mainly concentrated near the pattern parallel to the magnetic field, which represents the excitation of the propagation surface plasma resonance (PSPR) at the interface between the resonator and the dielectric. In Figure 7, for frequencies f_1_ and f_2_, since the energy is concentrated in the medium between the adjacent small square rings, the absorption is caused by the coupling of the adjacent square rings. In addition, there is a concentration of energy on the pattern parallel to the electric field for all resonant frequencies except for f_1_, so we believe that the electric dipole resonance is excited. According to the power distribution at seven absorption peaks, it can be concluded that the ultra-broadband absorption is caused by the combination of multiple resonance modes such as electric dipole resonance and PSPR.

Incident light is coupled into the resonator array, thereby creating surface currents that result in ohmic loss. We obtain the absorption of incident light by each part of the absorber through calculating the ohmic power loss [50]
(9)$$Q\left(\omega \right)=\frac{1}{2}\times \omega \times \varepsilon^{\prime \prime }\times {\left|E\left(\omega \right)\right|}^{2}$$
where *ω* is the angular frequency, *ε*″ is the imaginary part of the permittivity of the material, and *E*(*ω*) is the electric field corresponding to *ω*. In the simulation of COMSOL Multiphysics, the ratio of the integral of the ohmic power loss over the volume to the power of the simulated input port we set is the absorptivity of each part. The volume integral of different parts of the absorber can be calculated by calling the function and setting the domain. The calculation results are shown in Figure 8. Between 4 THz and 8 THz, the absorptance of the small square ring array is slightly higher than that of the large square ring array. For other frequency bands, the large ring array contributes significantly more to the absorption and reaches a maximum absorptance of 86.88% near 1.9 THz, which is consistent with the result of strong coupling between the large square rings at 1.93 THz in the field distribution. The absorptance of Topas is zero. We attribute the weak absorption in the ground plane to the intrinsic absorption of gold.

In general, the geometric parameters of the absorber will have some influence on the resonant frequency and absorption. During the design process of the absorber, extensive numerical simulations are required to optimize the geometric parameters. In Figure 9, we investigate the effect of increasing the thickness of VO_2_ from 0.05 μm to 0.25 μm on absorption. Figure 9a shows the simulation results of the absorptivity for different thicknesses of large square ring. It can be observed that with the increase of t_1_, the absorption bandwidth broadens, and both the low-frequency and high-frequency resonances are enhanced. The absorption becomes the largest when t_1_ = 0.1 μm. As can be seen in the discussion of the thickness of the small square ring in Figure 9b, the change in the intensity of the central absorption peak is obvious. The absorption is optimal at t_2_ = 0.15 μm. Additionally, the absorption bandwidth is hardly affected by t_2_. Combined with the extent to which the resonators contribute to the absorption in Figure 8, we can estimate the frequency range over which the geometric parameters have an influence on the absorption. We also show the effect of different opening sizes of the square ring on absorption in Figure 9. As shown in Figure 9c, it is clear that the absorption spectrum remains stable even though the actual size differs by more than 50% from our design. For the opening size of the small square ring, as shown in Figure 9d, it is interesting that larger openings and thinner thickness have similar effects. We think both are effects of changing the volume of the small square ring. Overall, the change of a single parameter has a limited impact on the absorption, and the absorption intensity in the bandwidth can be maintained above 80% in most cases. It will have important implications for practical fabrication.

In practical applications, the wide-angle absorption characteristics are of great significance to the measurement of the performance of the absorber. Figure 10 investigates the effect of different incident angles on the absorption performance, and the result shows that the absorption spectra of transverse electric (TE) polarization and transverse magnetic (TM) polarization are almost identical at small incidence angles. For TE polarization, as shown in Figure 10a, the lowest resonance frequency remains stable within a 40° span, while the high resonance appears blue-shifted, resulting in a slight increase in the bandwidth over 90% absorptance. As the incident angle further increases, the tangential component of the magnetic field decreases, and the absorptance continues to drop, resulting in a contraction of the absorption bandwidth. For TM polarization, as shown in Figure 10b, the absorption spectra look like those of TE polarization in the range of 40°. Before the incident angle increases to 65°, a continuum bandwidth of absorption greater than 90% can be maintained. Comparatively, deterioration of absorption due to higher-order diffraction at high incidence angles as in previous reports on absorbers [37,42] is not seen here. Obviously, our designed absorber can maintain superior absorption performance over a wide range of incident angles.

## 4. Conclusions

In conclusion, we have designed a tunable ultra-broadband terahertz absorber based on the VO_2_ and performed numerical simulations. The absorber consists of the metal ground plane and two-layer VO_2_ square ring arrays of different sizes surrounded by dielectric. The results show that the frequency bandwidth of over 90% absorptance reaches 10.76 THz from 1.63 THz to 12.39 THz, and the relative bandwidth is 153.5%, which indicates that the performance is greatly improved compared to the results of the previously reported absorbers based on VO_2_. By changing the conductivity of VO_2_ from 200 S/m to 2 × 10^5^ S/m, the absorption intensity can be dynamically adjusted in the range of 4.4–99.9%. We have explained the physical mechanism of complete absorption by impedance matching theory and power distribution and made absorption bandwidth and absorption efficiency optimal through adjusting geometric parameters. At the same time, it is proved that the absorption intensity is still within the acceptable range when the geometric parameters are slightly deviated. This has positive implications for practical fabrication. Moreover, our designed absorber has polarization insensitive characteristics because of its symmetry, and the absorption performance can be kept stable, when the incident angle varies up to 40° for TE polarization and 65° for TM polarization. Consequently, our designed absorber has great application prospects in modulation, stealth, optoelectronic switches, and other fields.

## Figures and Tables

**Figure 1 micromachines-13-00669-f001:**
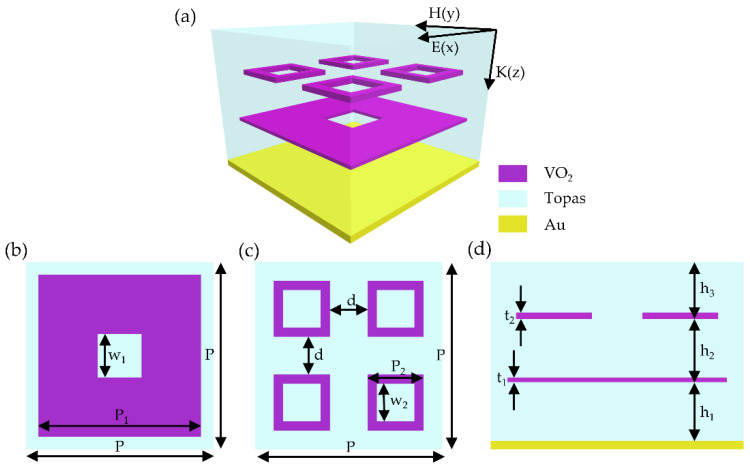
(**a**) The schematic of the unit cell of the proposed terahertz absorber. (**b**) The top view of the unit cell with large square ring. (**c**) The top view of the unit cell with small square rings. (**d**) The side view of the unit cell.

**Figure 2 micromachines-13-00669-f002:**
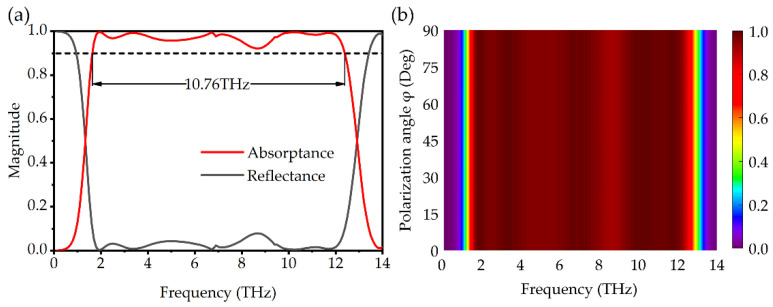
(**a**) The absorption and reflection spectra of the absorber. (**b**) The color map of the absorption spectra with different polarization angles.

**Figure 3 micromachines-13-00669-f003:**
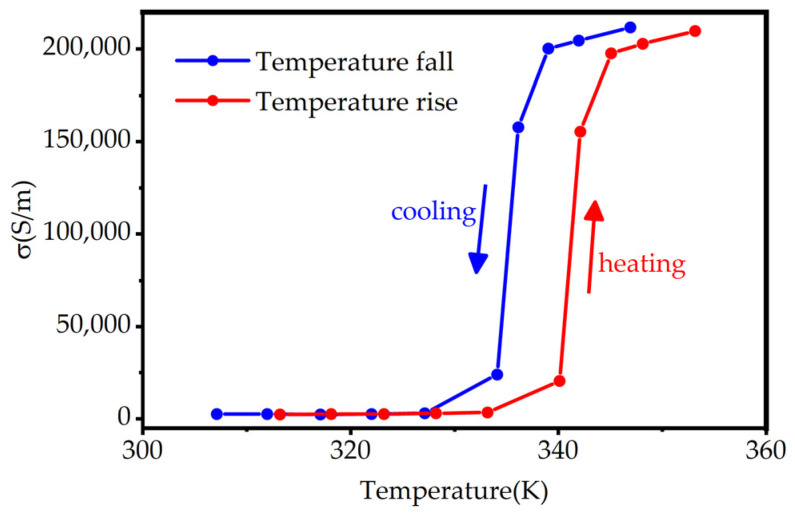
Schematic diagram of the relationship between VO_2_ conductivity and temperature.

**Figure 4 micromachines-13-00669-f004:**
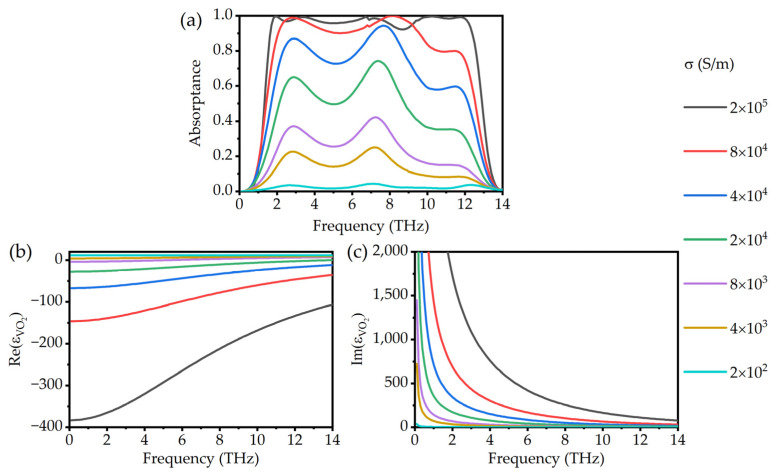
(**a**) The absorption spectra with different conductivities of VO_2_. (**b**) Real parts and (**c**) imaginary parts of permittivity with different conductivities of VO_2_.

**Figure 5 micromachines-13-00669-f005:**
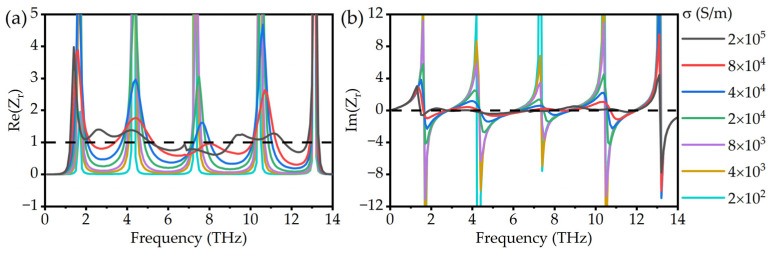
(**a**) Real parts and (**b**) imaginary parts of the relative impedance *Z_r_* with different conductivities of VO_2_.

**Figure 6 micromachines-13-00669-f006:**
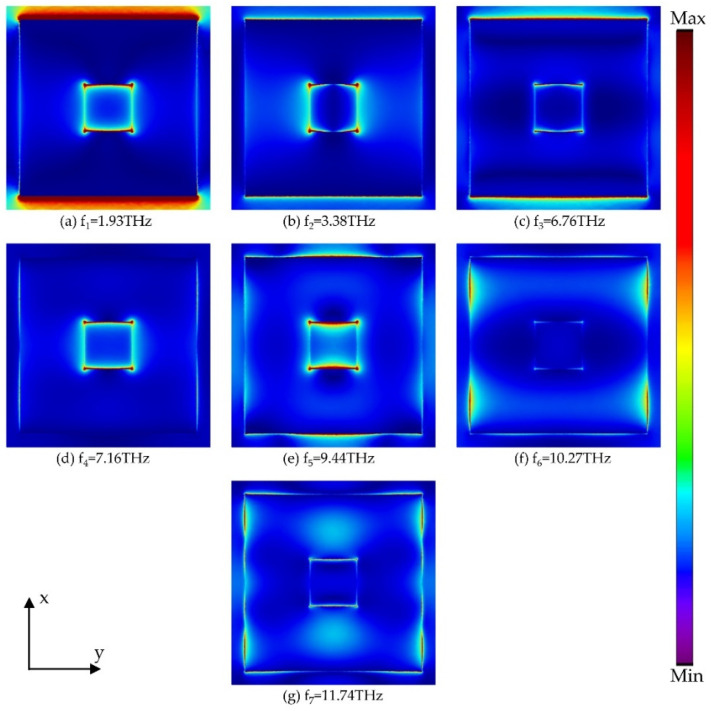
The power distribution of the large square ring at seven absorption peaks.

**Figure 7 micromachines-13-00669-f007:**
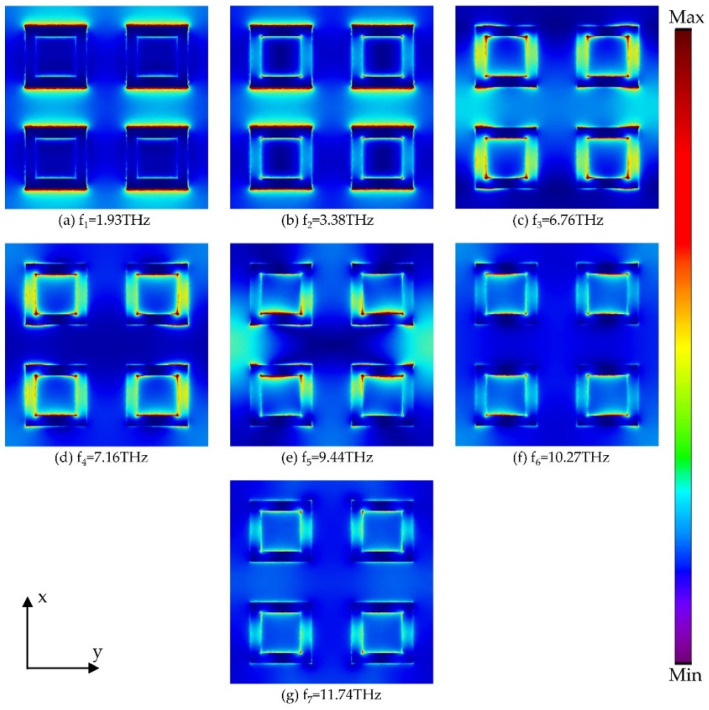
The power distribution of the small square ring at seven absorption peaks.

**Figure 8 micromachines-13-00669-f008:**
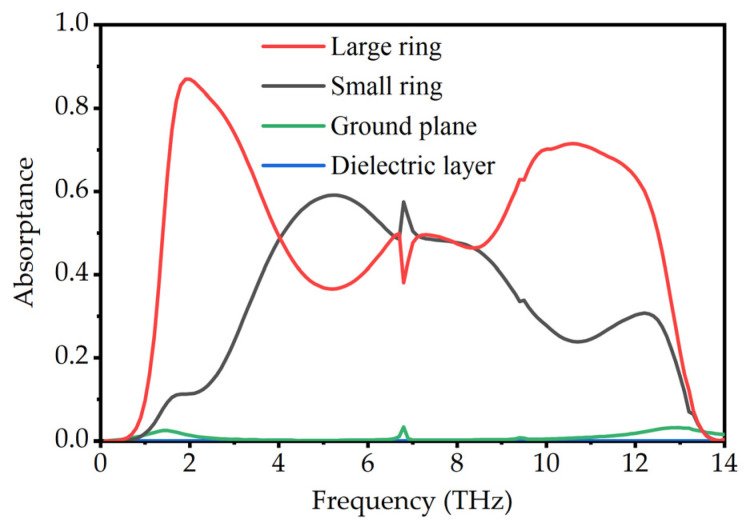
The absorptance of each component of the absorber.

**Figure 9 micromachines-13-00669-f009:**
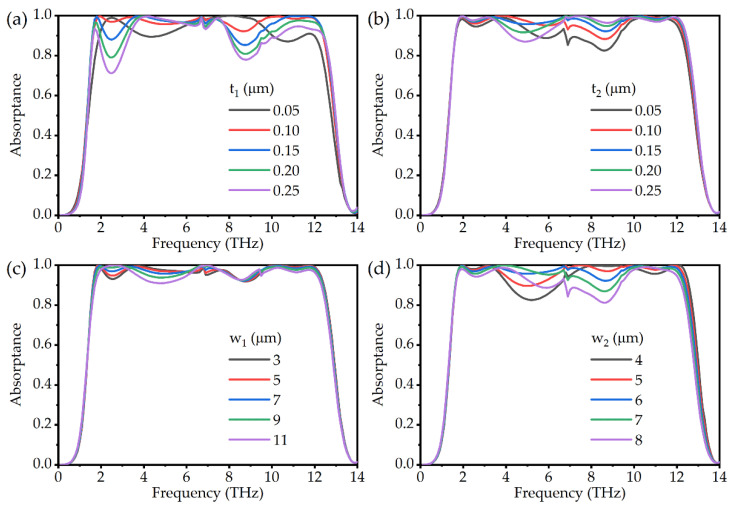
Absorption spectra of the absorber with different thicknesses of (**a**) large square ring and (**b**) small square ring, with different opening sizes of (**c**) large square ring and (**d**) small square ring.

**Figure 10 micromachines-13-00669-f010:**
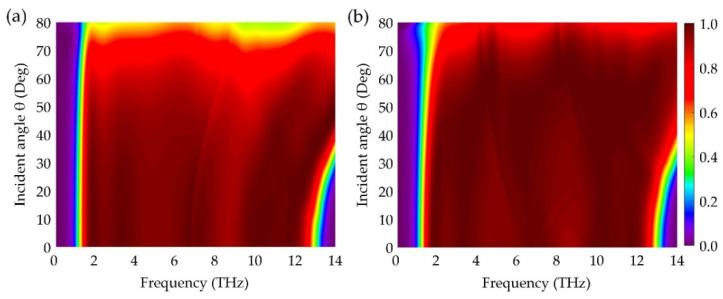
The absorption spectra with different incident angles for (**a**) TE polarization and (**b**) TM polarization.

**Table 1 micromachines-13-00669-t001:** Performance comparison between the newly designed absorber and the absorbers reported in recent years.

Reported Year and Reference	Absorption Bandwidth (THz)	Relative Bandwidth (%)	Tunable Range (%)	Design Features
2019 [35]	5.28 (10.28–15.56)	40.9	4.2–100	Combination of various patterns
2019 [36]	9.31 (7.36–16.67)	77.4	5.4–100	Stack of two multi-patterned layers
2020 [37]	2.45 (1.85–4.30)	79.7	4–100	Different spacing of patterns
2021 [38]	3.43 (0.93–4.36)	129.7	8–100	Patterns wrapped in dielectric
2021 [39]	3.30 (2.34–5.64)	82.7	4–100	Combination of different patterns
2021 [40]	4.66 (3.14–7.80)	85.2	2–99	Stack of two resonant layers
This work	10.76 (1.63–12.39)	153.5	4.4–99.9	Two layers wrapped in dielectric

## Data Availability

Research data presented in this study are available on request from the corresponding author.

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
