# Peer review of "Ultra-Broadband Tunable Terahertz Metamaterial Absorber Based on Double-Layer Vanadium Dioxide Square Ring Arrays"

_micromachines, 2022, doi:10.3390/mi13050669_

Round 1

Reviewer 1 Report

The work by Zhang et al. present a tunable terahertz absorber based on VO2 square ring arrays. The energy range and the absorption band width is quite wide making this work interesting. Also the possibility of tuning the absorption features is very valuable. I believe this work can be a great interest to the community and should be published, but some parts of the work should be improved before the publication.

  1. The used material VO2 is interesting and possess temperature induced phase transition as stated in the manuscript. Authors should add a reference for the general readers who might be interested with the properties of the material.
  2. Authors presented a table for the VO2 based tunable absorbers. What is the important parameter that makes these absorber quite different from each other? design? Why the authors way provide much larger tunable range. 
  3. Authors define the absorption features of VO2 and gold with Drude model. Where the variables like epsilon_infinity =12 is taken from? Also Drude model is for the itinerant systems, what happens when VO2 is an insulator? 
  4. Authors present different properties based on the different conductance of the VO2. So apparently conductance is the tuning parameter in this work, but there is no information given how the authors actually tune the conductivity? Gating? Temperature?
  5. Line 146 --> Absorptance perhaps should be corrected.
  6. In the last parts of the work, authors focus on the power distribution on the devices, and check the individual components and the absorptions therein. Is these are real measurements or simulations? In either case, how the individual components are obtained? 

Author Response

Response to Reviewer 1 Comments

The work by Zhang et al. present a tunable terahertz absorber based on VO2 square ring arrays. The energy range and the absorption band width is quite wide making this work interesting. Also the possibility of tuning the absorption features is very valuable. I believe this work can be a great interest to the community and should be published, but some parts of the work should be improved before the publication.

Point 1: The used material VO2 is interesting and possess temperature induced phase transition as stated in the manuscript. Authors should add a reference for the general readers who might be interested with the properties of the material.

Response 1: Thanks for your advice. We have added references to supplement the introduction to VO2 material properties. (ref.33 and ref.34)

Point 2: Authors presented a table for the VO2 based tunable absorbers. What is the important parameter that makes these absorber quite different from each other? design? Why the authors way provide much larger tunable range.

Response 2: We think that the key of improving absorber performance is the design of the structure. In previous reports, more resonator layers(ref.36,40) or more complex resonator patterns(ref.35,36) such as snowflakes and sawtooth can result in more absorption peaks and thus wider absorption bandwidth. In our design, we use simple square ring patterns and only two layers of resonators. Instead of designing the typical substrate-dielectric-resonator sandwich(ref.35,37,39) or simple stacks of dielectric and resonator layers(ref.36,40), we want to completely wrap the resonator in the dielectric. Such a design enhances the impedance matching between the absorber and free space for perfect absorption. And the decrease in the conductivity of VO2 will destroy the impedance matching conditions, resulting in strong reflection of electromagnetic waves. It makes our designed absorber maintaining excellent tuning capability while keeping a wide absorption bandwidth. In this regard, we have added a brief introduction to the absorber design in Table 1, so that the readers can more easily notice the differences between them.

Point 3: Authors define the absorption features of VO2 and gold with Drude model. Where the variables like epsilon_infinity =12 is taken from? Also Drude model is for the itinerant systems, what happens when VO2 is an insulator?

Response 3: The variables we used are all taken from the references we provide when introducing the Drude model. As shown in Figure 4 in the article, when VO2 is in the insulator phase, the real part of the relative permittivity of VO2 is about constant. And the imaginary part associated with the loss is extremely small. This is the reason that the absorptance is low when VO2 is in the insulating phase.

Point 4: Authors present different properties based on the different conductance of the VO2. So apparently conductance is the tuning parameter in this work, but there is no information given how the authors actually tune the conductivity? Gating? Temperature?

Response 4: It is a feasible way to control the conductivity of VO2 by adjusting the temperature. And the method of controlling VO2-based devices by electric heating has been proved feasible in practice [1]. In the article, we add Figure 3 to illustrate the relationship between temperature and VO2 conductivity and to represent the conditioning process. The mutation of conductivity occurs around 340K, and the whole process is reversible.

Point 5: Line 146 --> Absorptance perhaps should be corrected.

Response 5: Thanks for your reminder. Although we are sorry that we didn't find the mistake in line 146 of the original manuscript you describe, we have carefully checked the article and corrected the error found in Figure 2, which may be the same as you pointed out.

Point 6: In the last parts of the work, authors focus on the power distribution on the devices, and check the individual components and the absorptions therein. Is these are real measurements or simulations? In either case, how the individual components are obtained?

Response 6: It is the result of the simulation. We are sorry that the presentation in the article is not clear enough and we have improved our expression.(line191-195) The absorption of each part is calculated by the ratio of the integral of the ohmic power loss over the volume to the power of the simulated input port. In the simulation of COMSOL Multiphysics, the integral of the ohmic power loss over the volume of each part of the absorber can be calculated by calling the function and adding the domains in which the parts are located.

  1. Xiao, L.; Ma, H.; Liu, J.; Zhao, W.; Jia, Y.; Zhao, Q.; Liu, K.; Wu, Y.; Wei, Y.; Fan, S.; Jiang, K., Fast Adaptive Thermal Camouflage Based on Flexible VO2/Graphene/CNT Thin Films. Nano Lett. 2015, 15, 8365-8370.

Reviewer 2 Report

The manuscript ‘Ultra-broadband tunable terahertz metamaterial absorber 2 based on double-layer vanadium dioxide square ring arrays’ presents an interesting design of metasurface providing almost perfect absorption in wide spectral range. A lot of parameters are analyzed numerically to support high performance of the proposed metasurface. I think it makes sense to also estimate the influence of the non-perfect fabrication process, which could lead to deviation of metaatoms from the perfect geometry. The analysis of robustness of the metasurface vs fabrication uncertainties could help a lot in the experimental realization of the proposed metasurface.  

Author Response

Response to Reviewer 2 Comments

The manuscript ‘Ultra-broadband tunable terahertz metamaterial absorber 2 based on double-layer vanadium dioxide square ring arrays’ presents an interesting design of metasurface providing almost perfect absorption in wide spectral range. A lot of parameters are analyzed numerically to support high performance of the proposed metasurface. I think it makes sense to also estimate the influence of the non-perfect fabrication process, which could lead to deviation of metaatoms from the perfect geometry. The analysis of robustness of the metasurface vs fabrication uncertainties could help a lot in the experimental realization of the proposed metasurface. 

Response: Thank you for your suggestion. In order to better illustrate the robustness of metamaterials, we have added a study on the effect of the opening size of the square ring on the absorption in the study of geometric parameters.(line219-234) The results show that the opening size of the large square ring has little effect on the absorption performance, and the absorption intensity can be stabilized at more than 90%. For small square rings, the absorption intensity can be maintained above 80% if the opening size is within the tolerance of 2μm. In addition, the thickness study of vanadium dioxide also shows that the absorption intensity can be maintained above 80% in most cases even if the practical parameters deviate slightly from the design. We think that the possible effects of these errors are acceptable in fabrication. It has positive implications for the practical fabrication of the device.

Reviewer 3 Report

This paper presents an ultra-broadband tunable terahertz metamaterial absorber based on vanadium dioxide (VO2) material. Some comments are shown below.

  1. How to control the conductivity of the VO2 in the practical experiment?
  2. In Fig. 1, the thicknesses of two layers of VO2 are different. How to flexibly obtain the diverse thicknesses of VO2?
  3. Some recently reported works on VO2 based terahertz metamaterial absorbers should be cited to better understand the presented work, such as [R1] and [R2].

[R1] “Dynamically switchable terahertz absorber based on a hybrid metamaterial with vanadium dioxide and graphene,” J. Opt. Soc. Am. B, vol. 38, no. 11, pp. 3425-3434, 2021.

[R2] “Switchable and tunable bifunctional THz metamaterial absorber,” J. Opt. Soc. Am. B, vol. 39, no. 3, pp. A52-A60, 2022.

  1. For this kind of works, the key point is the realization of the experimental fabrication. Do the authors have any idea to solve the issue of the experimental fabrication?

Author Response

Response to Reviewer 3 Comments

This paper presents an ultra-broadband tunable terahertz metamaterial absorber based on vanadium dioxide (VO2) material. Some comments are shown below.

Point 1: How to control the conductivity of the VO2 in the practical experiment?

Response 1: In practical experiments, VO2-based devices can be regulated by electrical heating. We can control the conductivity of VO2 by placing the device on a heating plate and measuring the temperature with a thermocouple. Lin et al. have used this heating method in their experiments[1]. In the article, we supplement the Figure 3 to illustrate the relationship between temperature and VO2 conductivity. The sudden change in conductivity occurs around 340K and the process is reversible.

Point 2: In Fig. 1, the thicknesses of two layers of VO2 are different. How to flexibly obtain the diverse thicknesses of VO2?

Point 3: Some recently reported works on VO2 based terahertz metamaterial absorbers should be cited to better understand the presented work, such as [R1] and [R2].

[R1] “Dynamically switchable terahertz absorber based on a hybrid metamaterial with vanadium dioxide and graphene,” J. Opt. Soc. Am. B, vol. 38, no. 11, pp. 3425-3434, 2021.

[R2] “Switchable and tunable bifunctional THz metamaterial absorber,” J. Opt. Soc. Am. B, vol. 39, no. 3, pp. A52-A60, 2022.

Response 3: Thanks for the advice, as you said, these are novel works, and we have added these in the introduction.(line46-47)

Point 4: For this kind of works, the key point is the realization of the experimental fabrication. Do the authors have any idea to solve the issue of the experimental fabrication?

Response 2 and 4: The thickness of VO2 is determined by us through extensive numerical simulations. As shown in Figures 9a and 9b in the article, the absorber works best under this condition. But we are sorry, due to the consideration of experimental equipment and cost, we have no plans to prepare samples and conduct tests. We focus on providing theory and design guidance for a high-performance terahertz absorber. Nonetheless, we learn from the references that the preparation technology of VO2 thin film has matured[2-4]. Topas, often also called cyclic olefin copolymers (COC), is a relatively new class of polymer compounds. In recent years, solutions to the challenges of adhering resonators and ground planes on COC are proposed[5], such as oxygen plasma or UV treatment of COC and direct deposition of gold resonators on COC. Fabrication of COC-based multilayer devices[5]. Stage 1: First spin-coat PMMA on the substrate and dehydrate it at 120 °C. COC is then spin-coated on the PMMA/Si wafer. This is followed by patterning and deposition of the first layer of resonators. Stage 2: Stage 2 is conducted on a different wafer. First, COC is spin-coated on a silicon wafer coated with PMMA. Free-standing COC is released and attached to PDMS-coated silicon wafers. The role of PDMS is to facilitate release and is not affected by subsequent heat treatment. Stage 3: Next, place the COC/PDMS/Si wafer on the first layer of resonators. Bond using a wafer bonder at 100 °C, 400 N. The bonded COC can be easily released by peeling off the PDMS-coated wafer. The second layer of resonators is then similarly patterned and deposited. Repeating the above steps can fabricate a multilayer resonator COC-based device. We believe that a similar approach is possible to fabricate our designed absorber.

  1. Xiao, L.; Ma, H.; Liu, J.; Zhao, W.; Jia, Y.; Zhao, Q.; Liu, K.; Wu, Y.; Wei, Y.; Fan, S.; Jiang, K., Fast Adaptive Thermal Camouflage Based on Flexible VO2/Graphene/CNT Thin Films. Nano Lett. 2015, 15, 8365-8370.
  2. Huang, T.; Yang, L.; Qin, J.; Huang, F.; Zhu, X.; Zhou, P.; Peng, B.; Duan, H.; Deng, L.; Bi, L., Study of the phase evolution, metal-insulator transition, and optical properties of vanadium oxide thin films. Mater. Express 2016, 6, 3609-3621.
  3. Zhu, H.-F.; Du, L.-H.; Li, J.; Shi, Q.-W.; Peng, B.; Li, Z.-R.; Huang, W.-X.; Zhu, L.-G., Near-perfect terahertz wave amplitude modulation enabled by impedance matching in VO2 thin films. Phys. Lett. 2018, 112.
  4. Shi, Q.; Huang, W.; Zhang, Y.; Yan, J.; Zhang, Y.; Mao, M.; Zhang, Y.; Tu, M., Giant Phase Transition Properties at Terahertz Range in VO2 films Deposited by Sol-Gel Method. ACS Appl. Mater. Interfaces 2011, 3, 3523-3527.
  5. Ako, R. T.; Upadhyay, A.; Withayachumnankul, W.; Bhaskaran, M.; Sriram, S., Dielectrics for Terahertz Metasurfaces: Material Selection and Fabrication Techniques. Opt. Mater. 2020, 8.

Round 2

Reviewer 1 Report

The authors answered all the points raised in my previous report. I think this manuscript can be published as it is. 

Reviewer 2 Report

I am satisfied with your revision and think paper can be published a it is.

Reviewer 3 Report

The authors have addressed all my concerns.